# The hidden costs of inflation: A critical analysis of industrial development and environmental consequences

Dan Zheng[1], Abdullah Addas[2,3]*, Liaqat Ali Waseem[4], Syed Ali Asad Naqvi[4], Muneeb Ahmad[5], Kashif Sharif[6]

1 School of Law, Southwestern University of Finance and Economics, Chengdu, China, 2 Department of Civil Engineering, College of Engineering, Prince Sattam Bin Abdulaziz University, Al-Kharj, Saudi Arabia, 3 Landscape Architecture Department, Faculty of Architecture and Planning, King Abdulaziz University, Jeddah, Saudi Arabia, 4 Department of Geography, Government College University Faisalabad, Faisalabad, Punjab, Pakistan, 5 Department of Finance, Riphah International University, Islamabad, Pakistan, 6 Department of Statistics, University of Agriculture, Faisalabad, Pakistan

* a.addas@psau.edu.sa

**Data Availability Statement:** The data is taken from the World Bank database https://databank.worldbank.org/, World Economic Outlook Database (https://www.imf.org.com), and China Premium

## Abstract

The study draws attention to the associations between monetary and economic elements and their potential environmental impacts. The study uses time series data from 1960 to 2022 to examine the connection between $CO_2$ emissions, industrial growth, GNE, and inflation in China. The researchers utilized the well-known econometric technique of nonlinear autoregressive distributed lag (NARDL) to examine nonlinear correlations between these variables. The results reveal that GDP, inflation, and economic development influence long-term $CO_2$ emissions. The strong positive correlation between gross national expenditures and economic activity increases $CO_2$ emissions. In the short run, $CO_2$ emissions are positively and statistically significantly affected by inflation. While inflation temporarily affects $CO_2$ emissions, this effect dissipates with time. Industrial activity increases $CO_2$ emissions, and China's fast industrialization has damaged the environment. The energy-intensive fertiliser manufacturing process and fossil fuels increase $CO_2$ emissions. The research shows how government officials and academics may collaborate to create tailored measures to alleviate the environmental impacts of economic activity.

## 1. Introduction

Inflation indirectly leads to pollution in any country through increased production costs and altered consumption patterns. Government expenditures impact pollution through funding environmental regulations, green initiatives, infrastructure, and pollution cleanup efforts. While promoting economic growth, industrial development contributes to pollution through emissions, waste, land use changes, and deforestation. According to [1], gross national expenditures on R&D for technological innovation increase $CO_2$ emissions. Inflationary pressures have implications for various sectors of the economy, including energy production and

Database (https://www.ceicdata.com), National Bureau of Statistics of China (http://www.stats.gov.cn.com).

**Funding:** The authors extend their appreciation to Prince Sattam bin Abdulaziz University for funding this research work through the project number (PSAU/2023/01/8910, awarded to AA).

**Competing interests:** The authors have declared that no competing interests exist.

consumption. The inflation due to heavy expenses causes the excess use of fossil fuels, which results in the growth of $CO_2$ emissions [2]. Industrial activities are often associated with significant energy consumption and $CO_2$ emissions. By analysing the impact of industrial growth on $CO_2$ emissions, policymakers can identify strategies to strike a balance between economic development and environmental sustainability. Manufacturing agglomeration and total factor energy efficiency have a U-shaped connection with ecological systems [3]. The study objectives include exploring potential nonlinear and asymmetric effects among these variables. Low nutrient use efficiency has been associated with environmental degradation due to uneven fertiliser application and global usage [4]. The overuse of agrochemicals in agriculture degrades natural resources and threatens ecosystems [5]. The study aims to understand how inflation, national government expenditures, and industrial development increase or decrease environmental pollution in China, a fast-growing world economy. The research entails evaluating causation, recognising policy implications, and analysing data to guide sustainable behaviours. It's an interdisciplinary strategy that considers geographical and temporal differences while supporting sustainable development and lowering pollution. The study examined the effects of inflation, national government expenditures, and industrial development on China's carbon emissions from 1960 to 2022. The NARDL model is also used to account for projected asymmetries in stock market reactions to shifting inflation, national spending, and industrial development effects on environmental degradation. The study should distinguish between the effects of inflation and industrial growth on various forms of environmental pollution. Examining how inflation and economic development affect particular industrial sectors is crucial. It is important to consider how societal attitudes, consumer behaviour, and corporate accountability influence how inflation and industrial expansion affect environmental degradation. The study is exclusive in that it is the first to employ time-series analyses with long-range time-series data in a single country (China) from 1960 to 2022 to apply the ARDL and NARDL models. The results reveal a dynamic relationship between GNE, inflation, industrial growth, and $CO_2$ emissions. Inflation increases China's energy consumption, industrial growth, and transportation-related environmental pollution. The results show that government national expenditure affects $CO_2$ emissions, and the GNE mitigates climate change and reduces greenhouse gas emissions. The findings also show that China's $CO_2$ emissions have significantly grown due to industrial expansion, which greatly influences $CO_2$ emissions. Agricultural fertiliser consumption harms $CO_2$ emissions, so sustainable agricultural practices are essential. Promoting efficient nutrient management and reducing nutrient runoff also helps minimize fertiliser consumption's impact on environmental pollution. The study suggests that the industrial development of globalization and increased urbanisation have contributed to rising $CO_2$ emissions in China.

The subsequent portion of the investigation comprises four distinct phases. Following this, Section 2 provides an overview of the literature study. Section 3 delineates the research technique employed in the study, whereas Section 4 expounds on the outcomes and subsequent comments derived from the data analysis. Section 5 provides an overview of the results drawn from the study and offers ideas for further research and improvements.

## 2. Literature review

The US and German governments' national spending on encouraging R&D initiatives targets lowering $CO_2$ emissions [6]. An increase in government national expenditures significantly increases forest land clearing for agricultural production in the short run, leading to more $CO_2$ emissions [7]. The amount of money a country spends on R&D is correlated with how much carbon dioxide it produces [8]. Government national expenditures on research and

development to increase industrial growth and industrial development also increase environmental pollution [9]. According to [10], gross national expenditures on R&D investments may only sometimes lead to higher $CO_2$ emissions. [11] determined that the disparities in government national expenditures were a significant factor in national $CO_2$ emissions. Government national expenditures on renewable energy and exports have reduced $CO_2$ emissions [12]. The inflation rate and building material prices decline while $CO_2$ emissions grow [13]. According to [14], inflation considerably reduces GDP growth, which causes an increase in environmental pollution. According to [15], while more inflation is good for the government's gross domestic product, it also leads to more pollution as businesses and farms switch to cheaper fuel sources. There is no substantial correlation between inflation and renewable energy, but over the long run, renewable energy harms inflation in the short run [16]. The decrease in inflation rates has been increasing renewable energy sources, and the increase in renewable energy sources also decreases $CO_2$ emissions in the long run [17]. According to [18], the increase in the cost of renewable energy sources also decreases $CO_2$ emissions in the short and long run. According to [19], the increase in R&D expenditures for technological innovations also upset the $CO_2$ emissions in China. The use of fertilisers in agriculture production increases the quantity of $NO_2$ in soil and air, which is also a major source of environmental pollution [20]. Higher fertiliser consumption has boosted Nepal's short- and long-term carbon dioxide emission levels [21]. Using fertilisers in the Guangdong region of China has increased environmental pollution [22]. Fertiliser consumption in crop production is very important in increasing $CO_2$ emissions [23]. The increase in fertiliser consumption increases $CO_2$ emissions, and the increase in renewable energy sources due to urbanisation and industrial growth lowers $CO_2$ emissions in developed countries [24]. The continuous use of agricultural fertilisers in crop production creates soil degradation and air pollution [25]. Financial assistance for agriculture will significantly impact the use of chemical fertilisers and carbon emissions in the agricultural sector [26]. Overusing chemical fertilisers causes increased $CO_2$ emissions from agricultural output [27]. Fertiliser use and agricultural employment decreased $CO_2$ emissions in the long run [23]. Fertiliser consumption and livestock production significantly increased $CO_2$ emissions in the short and long run [28]. According to [29], ARDL econometrics support the EKC hypothesis and the long-run relationship between industrial expansion and $CO_2$ emissions. Modernising industry development increases $CO_2$ emissions due to higher energy demand and supply chain changes [30]. According to industrial growth, $CO_2$ emissions have a U-shaped connection in the short- and long-run [31]. Upgrading industrial structures reduces $CO_2$ emissions [32]. Energy consumption, urbanisation, and economic expansion boost Pakistan's $CO_2$ emissions from industrial development [33]. Industrial investment in China is the biggest cause of rising $CO_2$ emissions [34]. Industrial expansion and fossil fuel use are major contributors to regional $CO_2$ emissions [35]. Renewable energy, bootstrap autoregressive distributed lag testing (ARDL), and nonlinear ARDL enhance environmental quality in the short and long term [36]. The ARDL and NARDL symmetric analyses show that economic growth increases $CO_2$ emissions, but crude oil prices and FDI inflows increase $CO_2$ emissions [37]. The baseline ARDL and NARDL techniques used in the research revealed that economic growth impacts $CO_2$ emissions [38]. According to [39], the ARDL and NARDL models determined that a rise in economic growth would reduce $CO_2$ emissions, while a decrease in economic growth would raise $CO_2$ emissions. [40] has applied the NARDL model and determined that population density and GDP per capita increase carbon emissions in the short and long run, while income inequality does not impact carbon emissions in the short run. In the ARDL model, economic growth increases energy consumption, and urbanisation increases $CO_2$ emissions [41].

**Table 1. Displays the variables' detail.**

| Variable | Description | Unit |
|----------|-------------|------|
| $CO_2$ | Carbon dioxide emissions | annual per capita |
| IN | Inflation | Million Dollars |
| GNE | Gross national expenditure | (% of GDP) |
| IG | Industry (including construction) | (annual % growth) |
| FR | Fertiliser consumption | (kilos per arable hectare) |

## 3. Research methodology

### 3.1 Data collection

The study has applied data from 1960 to 2022, which has been collected from the World Bank database "https://databank.worldbank.org.com" URL https://databank.worldbank.org/indicator/NY.GDP.MKTP.KD.ZG/1ff4a498/Popular-Indicators" for $CO_2$ emissions, inflation (IN), and fertiliser consumption (FR). The data for gross national expenditure (GNE) and industrial growth (IG) has been collected from the International Monetary Fund Database ("https://www.imf.org.com")," URL "https://climatedata.imf.org/pages/access-data" the World Economic Outlook and China Premium Database ("https://www.ceicdata.com)" URL "https://info.ceicdata.com/en/en-products-global-database.". China is a very fast-growing economy in the world, and due to its high industrial and agricultural production levels, it is the largest producer of $CO_2$ emissions. China's $CO_2$ emissions have a significant impact on global climate change. Understanding the factors contributing to these emissions is critical for addressing environmental concerns. China's economy is closely linked to its industrial growth and gross national expenditures. Understanding the relationship between these factors and $CO_2$ emissions can provide insights into the potential economic impacts of efforts to reduce emissions. Table 1 demonstrates the variables' symbols, units, and descriptions.

Understanding the global economic and environmental difficulties requires examining the connection between China's Gross National Expenditures, inflation, industrial expansion, and CO2 emissions.

### 3.2 Research design

Designing a clean and sustainable energy system is a complex task that requires a multidisciplinary approach, and the investigation uses the theoretical links to come up with the equation:

$$lnCO_2 = \alpha_0 + \alpha_1 lnIN_t + \alpha_2 lnGNE_t + \alpha_3 lnIG_t + \alpha_4 lnFP_t + \alpha_5 lnFR_t \quad (1)$$

The study employed $CO_2$ emissions as the dependent variable, whereas the independent variables are inflation, gross national product (GNE), industrial development, and fertilizer consumption. The research anticipates the following connections between these variables:

$\frac{PlnCO_{2t}}{PlnIN_t} < 0$ More pronounced degrees of ecological innovation correlate with lower fossil fuel by-products.

$\frac{PlnCO_{2t}}{PlnGNE_t} < 0$ Greater GNE results in higher $CO_2$ emissions.

$\frac{PlnCO_{2t}}{PlnFP_t} < 0$ The story of $CO_2$ emissions increases when FP levels rise.

$\frac{PlnCO_{2t}}{PlnIG_t} < 0$ IG use increases are correlated with decreased $CO_2$ emissions.

$\frac{PlnCO_{2t}}{PlnFR_t} < 0$ The smaller the $CO_2$ emissions, the higher the FR.

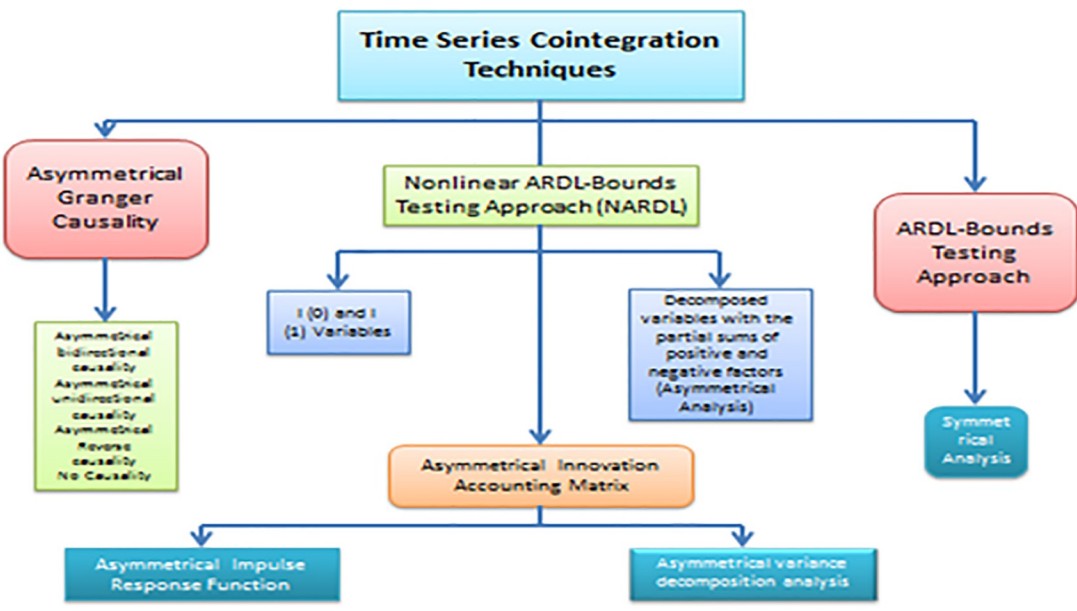

**Fig 1. Illustrates the graphical framework of the research methodology.**

## 3.3 Econometrical background

The NARDL model's graphical structure looks like this: Using standardized unit root tests, such as the Augmented Dickey-Fuller (ADF) test, establishes the order of integration of the variables. The generated graphical framework illustrates the causal connections between the NARDL model's variables. Fig 1 demonstrates the graphical framework of the nonlinear short ARDL-bounds testing approach (NARDL).

The NARDL approach allows both short-run and long-run dynamics in the data and allows nonlinear shorts to be captured in the data-generating process. To use the NARDL method, you have to estimate an autoregressive distributed lag (ARDL) model with lagged values for the dependent and independent variables and lagged differences between the variables. According to [42], a short- and long-run nexus between industrial growth and $CO_2$ emissions was identified using the ARDL bounds testing model. The significant difference between the traditional ARDL model and the NARDL model is the inclusion of a lagged squared term of the dependent variable in the model.

### Phase 1: Unit root test

The Augmented Dickey-Fuller (ADF) test has been used to examine the impact of the unit root of gross national expenditures (GNE), inflation, fertiliser, and industrial growth on $CO_2$ emissions. ADF tests the null hypothesis that a time series has a unit root to determine its non-stationarity. The null hypothesis is rejected, and the time series is considered stationary in Eq 2:

$$A_t = \varphi A_{t=1} + \epsilon_t \tag{2}$$

The series is viewed as fixed if the coefficient of the above condition approaches one, as shown by the AR (1) process $A_{t-1}$ is less than 1. If $|\varphi| < 1$, the data is stationary at level 1, and if $|\varphi| = 1$, the variable is non-stationary at the level and includes the unit root process. In the

third scenario, $|\varphi| > 0$ for the series never achieves equilibrium because it is diverse and explosive.

## Phase 2: ARDL model

The ARDL (Autoregressive Distributed Lag) model is a popular econometric approach used to analyze the long-term relationships among variables. The study has applied ARDL model to analyze the influence of gross national expenditures, inflation, fertilizer consumption, and industrial growth on $CO_2$ emissions. The ARDL model regression has represented as in Eq 3:

$$\Delta CO_{2t} = \beta_0 + \sum_{j=1}^{a} \beta_{1j} \Delta CO_{2t-1} + \sum_{j=1}^{b} \beta_{2j} \Delta IN_t + \sum_{j=1}^{c} \beta_{3j} \Delta FR_t +$$
$$\sum_{j=1}^{d} \beta_{4j} \Delta GNE_t + \sum_{j=1}^{e} \beta_{5j} \Delta FP_t + \sum_{j=1}^{f} \beta_{5j} \Delta IG_t + \gamma_1 CO_{2t-1} +$$
$$\gamma_2 IN_t + \gamma_3 FR_t + \gamma_4 GNE_t + \gamma_5 FP_t + \gamma_6 IG_t + \epsilon_t \tag{3}$$

Factor cointegration is optional, and F-statistics evaluate the hypothesis. Suppose the F-test result is greater than the factors cointegration. SBC and AIC will resolve slack time, and the long-run Eq 4 is as follows:

$$CO_{2t} = \gamma_1 CO_{2t-1} + \gamma_2 IN_t + \gamma_3 FR_t + \gamma_4 GNE_t + \gamma_5 FP_t + \gamma_6 IG_t + \epsilon_t \tag{4}$$

The short-run equation will be approximated if a long-run connection between these variables exists. The error correction run will be negative and demonstrate convergence to equilibrium.

$$\Delta CO_{2t} = \beta_0 + \sum_{j=1}^{a} \beta_{1j} \Delta CO_{2t-1} + \sum_{j=1}^{b} \beta_{2j} \Delta IN_t + \sum_{j=1}^{c} \beta_{3j} \Delta FR_t +$$
$$\sum_{j=1}^{d} \beta_{4j} \Delta GNE_t + \sum_{j=1}^{e} \beta_{5j} \Delta FP_t + \sum_{j=1}^{f} \beta_{5j} \Delta IG_t + \epsilon_t \tag{5}$$

## Phase 3: NARDL model

The study used the [43] NARDL model to detuning how each response variable responds to both positive and negative shocks and evaluate variable asymmetry in the short- and long-run. NARDL specifies the following equation in short- and long-run asymmetries:

$$CO_{2t} = \alpha_0 + \alpha_1 IN_t^+ + \alpha_2 IN_t^- + \alpha_3 FR_t^+ + \alpha_4 FR_t^- + \alpha_5 GNE_t^+ + \alpha_6 GNE_t^- +$$
$$\alpha_7 FP_t^+ + \alpha_8 FP_t^- + \alpha_9 IG_t^+ + \alpha_{10} IG_t^- + \nu_t \tag{6}$$

Where $\alpha_0, \alpha_1 \ldots \alpha_9$ are a set of estimated long-run parameters and based on above the preparation mentioned by '$\gamma$' lessening in $CO_2$ emissions. The supplementary coefficients that designate the particular connection among the NARDL model and other variables as:

$$\Delta CO_{2t} = \beta_0 + \sum_{K=0}^{p} \beta_1 \Delta CO_{2t-K} + \sum_{K=0}^{p} \beta_2 IN_{t-K}^+ + \sum_{K=0}^{p} \beta_2 IN_{t-K}^- + \sum_{K=0}^{p} \beta_2 FR_{t-K}^+ +$$
$$\sum_{K=0}^{p} \beta_2 FR_{t-K}^- + \sum_{K=0}^{p} \beta_2 GNE_{t-K}^+ + \sum_{K=0}^{p} \beta_2 GNE_{t-K}^- + \sum_{K=0}^{p} \beta_2 FP_{t-K}^+ + \sum_{K=0}^{p} \beta_2 FP_{t-K}^- +$$
$$\sum_{K=0}^{p} \beta_2 IG_{t-K}^+ + \sum_{K=0}^{p} \beta_2 IG_{t-K}^- + \gamma_1 CO_{2t-K} + \gamma_2 IN_{t-K}^+ + \gamma_2 IN_{t-K}^- + \gamma_2 FR_{t-K}^+ +$$
$$\gamma_2 FR_{t-K}^- + \gamma_2 GNE_{t-K}^+ + \gamma_2 GNE_{t-K}^- + \gamma_2 FP_{t-K}^+ + \gamma_2 FP_{t-K}^- + \gamma_2 IG_{t-K}^+ + \gamma_2 IG_{t-K}^- + \nu_t \tag{7}$$

The following describes the ECM examines the consistency of long-run parameters and the short- run consequences:

$$\Delta CO_{2t} = \beta_0 + \sum_{K=0}^{p} \beta_1 \Delta CO_{2t-K} + \sum_{K=0}^{p} \beta_2 IN_{t-K}^{+} + \sum_{K=0}^{p} \beta_2 IN_{t-K}^{-} + \sum_{K=0}^{p} \beta_2 FR_{t-K}^{+} +$$

$$\sum_{K=0}^{p} \beta_2 FR_{t-K}^{-} + \sum_{K=0}^{p} \beta_2 GNE_{t-K}^{+} + \sum_{K=0}^{p} \beta_2 GNE_{t-K}^{-} + \sum_{K=0}^{p} \beta_2 FP_{t-K}^{+} + \sum_{K=0}^{p} \beta_2 FP_{t-K}^{-} + (8)$$

$$\sum_{K=0}^{p} \beta_2 IG_{t-K}^{+} + + \sum_{K=0}^{p} \beta_2 IG_{t-K}^{-} + \pi_0 \text{ECT} + v_t$$

The NARDL model follows the stages and $\gamma_1 = \gamma^+ = \gamma^- = 0$, presents a different theory proposes that cointegration exists in long-run. If it falls in the range between the upper and lower boundaries represent the difficulty in making a choice and is referred to as another cointegration method. The following hypothesis is being tested by the Wald test, i.e., $\gamma_2^+ = \gamma_3^- = 0$ or $-\gamma_2^+/\gamma_1 = \gamma_3^-/\gamma_1$.

## Phase 4: The granger-toda-yamamoto principle

Toda-Yamamoto model dynamic VAR(p) model is written as:

$$
\begin{bmatrix} CO_{2t-1}^{+} \\ IN_{t-1}^{+} \\ FR_{t-1}^{+} \\ GNE_{t-1}^{+} \\ FP_{t-1}^{+} \\ IG_{t-1}^{+} \end{bmatrix} = \begin{bmatrix} \alpha \\ \beta \\ \gamma \\ \rho \\ \delta \\ \partial \end{bmatrix} + \sum_{j=1}^{p} \begin{bmatrix} P_{11j} & P_{12j} & P_{13j} & P_{14j} & P_{15j} & P_{16j} \\ P_{21j} & P_{22j} & P_{23j} & P_{24j} & P_{25j} & P_{26j} \\ P_{31j} & P_{32j} & P_{33j} & P_{34j} & P_{35j} & P_{36j} \\ P_{41j} & P_{42j} & P_{43j} & P_{44j} & P_{45j} & P_{46j} \\ P_{51j} & P_{52j} & P_{53j} & P_{54j} & P_{55j} & P_{56j} \\ P_{61j} & P_{62j} & P_{63j} & P_{64j} & P_{65j} & P_{66j} \end{bmatrix} \times \begin{bmatrix} CO_{2t-1}^{+} \\ IN_{t-1}^{+} \\ FR_{t-1}^{+} \\ GNE_{t-1}^{+} \\ FP_{t-1}^{+} \\ IG_{t-1}^{+} \end{bmatrix}
$$

$$
+ \sum_{i-p1}^{dmax} \begin{bmatrix} P_{11j} & P_{12j} & P_{13j} & P_{14j} & P_{15j} & P_{16j} \\ P_{21j} & P_{22j} & P_{23j} & P_{24j} & P_{25j} & P_{26j} \\ P_{31j} & P_{32j} & P_{33j} & P_{34j} & P_{35j} & P_{36j} \\ P_{41j} & P_{42j} & P_{43j} & P_{44j} & P_{45j} & P_{46j} \\ P_{51j} & P_{52j} & P_{53j} & P_{54j} & P_{55j} & P_{56j} \\ P_{61j} & P_{62j} & P_{63j} & P_{64j} & P_{65j} & P_{66j} \end{bmatrix} \times \begin{bmatrix} CO_{2t-1}^{+} \\ IN_{t-1}^{+} \\ FR_{t-1}^{+} \\ GNE_{t-1}^{+} \\ FP_{t-1}^{+} \\ IG_{t-1}^{+} \end{bmatrix} + \begin{bmatrix} v_1 \\ v_2 \\ v_3 \\ v_4 \\ v_5 \\ v_6 \end{bmatrix} \quad (9)
$$

Granger causality from GNE to $CO_2$ implies that $P_{11j} = 0$, and Granger basis from $CO_2$ to GNE implies that $P_{66j} = 0$. After looking at both short-run and long-run parameter estimates, the study used the Granger causality test to determine the hypotheses. Vector auto-regression (VAR) and vector error-correction models (VECM) were used to find the direction of Granger causality between GNE and $CO_2$ emissions over time. These models have been used to analyze the dynamic relationship between all the variables over time and test for the presence of Granger causality.

## Phase 5: Innovation accounting matrix

According to [44], IRF measures the dynamic interaction between the relevant variables across time and converts the VAR model. The primary application of VDA is to clarify the relative importance of variables. The Innovation Accounting Matrix has been applied to track and

measure progress in innovation related to the influence of gross national expenditures, Inflation, Fertilizer Consumption, and Industrial Growth on $CO_2$ Emissions. Gross National Expenditures, Inflation, Fertilizer Consumption, and Industrial Growth are typically used as economic indicators to measure a country's economic performance. $CO_2$ emissions measure the amount of carbon dioxide emissions into the atmosphere.

# 4. Results and discussions

## 4.1 Descriptive statistics

Table 2 demonstrates the descriptive statistics for $CO_2$ emissions, fertilizer consumption (FP) (percentage of output), and fertilizer consumption (FR) (kg per hectare of arable land). Descriptive statistics show that inflation (IN) and gross national expenditures (GNE) achieved their highest levels. The distribution of industrial growth (IG) (annual growth) as a percentage of overall GDP growth is adversely skewed. The study has a sample of data on $CO_2$ emissions, fertilizer consumption (FP), and fertilizer consumption (FR).

While overall GDP is growing, the industrial sector is growing at a different pace or may even be contracting. The interpretation of the distribution's means, median and skewness depends on the specific context and the goals of the analysis. In some cases, an adversely skewed distribution may be desirable, such as when trying to reduce economic inequality and more sustainable economic growth. A more symmetrical distribution is preferable, aiming for balanced economic growth across all sectors. Table 3 demonstrates the 1st difference level of IG, FP, and GNE non-stationary. In time series analysis, the factors' stationarity is essential because the non-stationarity of a unit root in the factors prompts erroneous relapse examinations.

The results demonstrate that the data contain a unit root; the IG is non-stationary and may show a trend or drift over time. FP and IN exhibit non-stationary unit roots, indicating temporal drift. GNE data has a trending non-stationary unit root, and the non-stationary ADF test shows a unit root for $CO_2$ emissions. Rising oil prices cut $CO_2$ emissions, limit high-carbon energy use, increase renewable power demand, and lower real GDP [45]. The investigation may proceed to the ARDL and NARDL testing procedures given the ARDL test conditions, no second request coordinated, or I (2) variable is found. An acceptable VAR lag period is necessary before employing the ARDL bound testing approach. Table 4 demonstrates an acceptable

**Table 2. Descriptive statistics.**

|  | $CO_2$ | FP | FR | GNE | IN | IG |
|---|---|---|---|---|---|---|
| Mean | 2.969274 | 125.3028 | 226.3129 | 97.83130 | 3.543775 | 9.794117 |
| Median | 2.172847 | 124.1269 | 213.6762 | 97.61566 | 2.054921 | 10.33835 |
| Maximum | 7.554165 | 200.5952 | 464.7764 | 104.1350 | 20.61699 | 34.60000 |
| Minimum | 0.604760 | 85.21471 | 7.040823 | 91.47848 | -3.792529 | -41.9 |
| Std. Dev. | 2.238031 | 25.26849 | 150.6972 | 2.622271 | 4.803005 | 10.84207 |
| Skewness | 0.921555 | 0.699893 | 0.063343 | 0.106729 | 1.363304 | -1.744982 |
| Kurtosis | 2.428874 | 3.746415 | 1.649781 | 3.354845 | 4.939297 | 10.79389 |
| Jarque-Bera | 9.308092 | 6.291344 | 4.597853 | 0.428699 | 27.98817 | 182.3115 |
| Probability | 0.009523 | 0.043038 | 0.100367 | 0.807066 | 0.000001 | 0.000000 |
| Sum | 178.1564 | 7518.169 | 13578.77 | 5869.878 | 212.6265 | 587.6470 |
| Sum Sq. Dev. | 295.5181 | 37671.29 | 1339868. | 405.7019 | 1361.062 | 6935.475 |
| Observations | 60 | 60 | 60 | 60 | 60 | 60 |

**Table 3. Demonstrates the unit root test results.**

| Variables | Augmented Dickey-Fuller | | | |
| --- | --- | --- | --- | --- |
| | Level | | 1st difference | |
| | C | C & T | C | C & T |
| $CO_2$ | 0.994861 | -1.153847 | -3.777592*** | -4.126648*** |
| IN | -3.842033*** | -3.884584** | -8.432128*** | -8.348631*** |
| GNE | -3.279960** | -3.179218* | -9.115068*** | -9.153441*** |
| IG | -5.501631*** | -5.459791*** | -9.854713*** | -9.900273*** |
| FP | -2.592418 | -3.502822** | -5.336529*** | -7.641984*** |
| FR | -1.037703 | -2.477468 | -6.044263*** | -5.986559*** |

Note:

***, **, and * denote significance at 1%, 5%, and 10%, respectively.

lag length for the VAR Lag Order Selection Criteria. Vector Auto-regression (VAR) models have been used to model the dynamic relationship between the study time series variables.

The AIC is a measure of the relative quality of a statistical model for the set of data. The BIC is similar to the AIC but places a higher penalty on models with other parameters. The HQIC is a modification of the AIC that provides a more accurate estimate of the optimal lag order for VAR models. The FPE is another measure of the quality of a statistical model that penalizes models with other parameters.

## 4.2 Wald test parameters

Table 5 demonstrates the Wald test results, a statistical test used to determine whether a group of parameters in a regression model are jointly significant. In the context of a VAR model, the Wald test has been used to test the joint significance of the coefficients of multiple lagged values of different time series variables, including Gross National Expenditures (GNE), inflation (IN), fertilizer consumption (FP), and industrial growth (IG), on $CO_2$ emissions.

The results demonstrated the influence of GNE, FR, IG, IN, and IG on $CO_2$ emissions. The Wald test has been used for the joint significance of coefficients in the ARDL and NARDL models. To perform the Wald test, the study first estimates the ARDL model. The study used the Wald test statistic to calculate the p-value for the test, and the p-value is less than a chosen significance level of 5%. Where C is a vector of the estimated coefficients of the interaction terms, and R is a matrix of the estimated covariance of the coefficients of the interaction terms. According to [46], government expenditures on R&D for renewal energy cause a decrease in $CO_2$ emissions, but due to high expenditures, Chinese companies are still using fossil fuels,

**Table 4. Demonstrates the VAR lag order selection criteria.**

| Lag | LogL | LR | FPE | AIC | SC | HQ |
| --- | --- | --- | --- | --- | --- | --- |
| 0 | - 1083.228 | NA | 2.63e+10 | 38.18344 | 38.36265 | 38.25309 |
| 1 | - 882.0448 | 360.0121 | 54630342 | 32.00157 | 33.07686* | 32.41947* |
| 2 | - 848.6623 | 53.88045 | 41556808 | 31.70745 | 33.67882 | 32.47359 |
| 3 | -818.1374 | 43.91304* | 36093647* | 31.51359* | 34.38103 | 32.62798 |

Note:

* indicates lag order selected by the test statistic (each test at 5% level).

**Table 5. Demonstration of wald test parametric statistical analysis.**

| Variable | Coefficient | Std. Error | t-Statistic | Prob.* |
|---|---|---|---|---|
| C | 0.480094 | 0.152965 | 3.138589 | 0.0032 |
| $CO_2$ (-1) | -0.10848 | 0.028087 | -3.86226 | 0.0004 |
| FP_POS | -0.000469 | 0.002065 | -0.22694 | 0.8216 |
| FP_NEG | -0.007933 | 0.001814 | -4.3728 | 0.0001 |
| FR_POS(-1) | 0.003399 | 0.000782 | 4.347691 | 0.0001 |
| FR_NEG | -0.003539 | 0.001099 | -3.21929 | 0.0026 |
| GNE_POS | 0.022558 | 0.015895 | 1.419177 | 0.1636 |
| GNE_NEG | -0.026989 | 0.014107 | -1.91325 | 0.0629 |
| IN_POS(-1) | -0.040576 | 0.008419 | -4.81965 | 0.0000 |
| IN_NEG(-1) | 0.004595 | 0.00457 | 1.005373 | 0.3208 |
| IG_POS(-1) | -0.007562 | 0.002666 | -2.83704 | 0.0071 |
| IG_NEG | 0.00413 | 0.002206 | 1.872328 | 0.0685 |
| D(CO2(-1)) | 0.388176 | 0.095041 | 4.084311 | 0.0002 |
| D(IN_POS (-1)) | 0.044188 | 0.009312 | 4.745145 | 0.0000 |
| D(IN_NEG (-1)) | -0.022127 | 0.007263 | -3.04663 | 0.0041 |
| D(FR_POS (-2)) | 0.002054 | 0.001071 | 1.917071 | 0.0624 |
| R-squared | 0.807834 | Mean dependent var | | 0.122969 |
| Adjusted R-squared | 0.735772 | SD dependent var | | 0.166027 |
| SE of regression | 0.085343 | Akaike info criterion | | -1.84932 |
| Sum squared residual | 0.291336 | Schwarz criterion | | -1.27065 |
| Log-likelihood | 67.78103 | Hannan-Quinn criter. | | -1.62497 |
| F-statistic | 11.21025 | Durbin-Watson stat | | 1.981584 |
| Prob(F-statistic) | 0 | | | |

*Note: p-values and subsequent tests do not account for stepwise, and p-value < 0.05.

which increases the $CO_2$ emissions. The Wald test further demonstrated that ARDL and NARDL models depict relationships between data properties.

## 4.3 ARDL model and bounds test

Table 6 demonstrates the [47] ARDL model in various orders, I (0) and I (1), in the short run. The ARDL model can handle data that are stationary in any order (I(0) or I(1)). The study has applied the [47] Autoregressive Distributed Lag (ARDL) model of order (2, 0, 3, 1, 1, 1) to estimate the short-run and long-run effects of independent variables (FP, FR, IG, IN and GNE) on a dependent variable ($CO_2$ emissions).

The study has analyzed the coefficients of FP, FR, IG, IN, and GNE influence $CO_2$ emissions. The results look at the sign and magnitude of each coefficient of independent variables FP, IG, IN and GNE statistical significance (using t-tests or p-values) to see whether these variables positively affect $CO_2$ emissions. The mechanism's effectiveness in promoting sustainable corporate development; however, implementing the mainland-HK Stock Connect has primarily boosted the leading firms [48]. Table 7 demonstrates the ARDL model's long-run assessments with the bound test.

The long-run coefficients provide information on the long-term impact of changes in the FP, FR, IG, IN, and GNE influences on $CO_2$ emissions. The long-run assessments with the bound test involve examining the long-run coefficients and statistical significance. A statistically significant coefficient reveals the long-term influence of FP, FR, IG, IN, and GNE on $CO_2$

**Table 6. Displays the ARDL model estimations for all variables in the short run.**

| Variables | Coeff. | Std. Er. | t-Stat. | Prob.* |
|---|---|---|---|---|
| $CO_2$ (-1) | 1.493646 | 0.126344 | 11.82208 | 0.0000 |
| $CO_2$ (-2) | -0.499176 | 0.127981 | -3.900407 | 0.0003 |
| FP | 4.05E-05 | 0.001348 | 0.030003 | 0.9762 |
| FR | 0.000178 | 0.001038 | 0.171275 | 0.8648 |
| FR (-1) | -0.000143 | 0.001581 | -0.090329 | 0.9285 |
| FR (-2) | 3.47E-05 | 0.001552 | 0.02234 | 0.9823 |
| FR (-3) | -0.002237 | 0.001249 | -1.791085 | 0.0805 |
| GNE | -0.003286 | 0.009904 | -0.331805 | 0.7417 |
| GNE (-1) | -0.012136 | 0.009602 | -1.263852 | 0.2133 |
| IN | 0.007327 | 0.00536 | 1.366957 | 0.1789 |
| IN (-1) | -0.007824 | 0.005091 | -1.536609 | 0.1319 |
| IG | 0.00531 | 0.002103 | 2.525392 | 0.0154 |
| IG (-1) | -0.004265 | 0.002021 | -2.110692 | 0.0408 |
| C | 1.359161 | 0.856294 | 1.58726 | 0.1200 |
| @TREND | 0.021916 | 0.008855 | 2.474915 | 0.0174 |
| R-squared | 0.998146 | Mean dependent var | | 3.076718 |
| Adjusted R-squared | 0.997528 | SD dependent var | | 2.244885 |
| SE of regression | 0.111625 | Akaike info criterion | | -1.326412 |
| Sum squared reside | 0.523325 | Schwarz criterion | | -0.788767 |
| Log-likelihood | 52.80273 | Hannan-Quinn criter. | | -1.117465 |
| F-statistic | 1614.806 | Durbin-Watson stat | | 1.921669 |
| Prob(F-statistic) | 0.00000 | | | |

Note: P-values are significant at 1%, 5% and 10%.

emissions. The government expenditures on energy technologies that meet the electricity demand, considering economic and environmental parameters [49]. The environment's quality is influenced over the long run by IG, GNE, FR, and FP impacts on $CO_2$ emissions are significantly and negatively impacted by IN, GNE, and FP, although IN and FP directly correlate to $CO_2$ emissions. The industrial and resource curse concept is supported by a country's higher GNE and IG and increased loss of natural resources, and China is eventually facing environmental pollution. Most nitrogen is lost to the environment, particularly soil, water, and air, leading to non-point source pollution [50]. Table 8 exhibits ARDL model error correction regression. Error correction evaluates the deviation from the long-run equilibrium relationship between GNE, IG, FR, IN, FP, and $CO_2$ emissions.

The results demonstrate that the GNE, IG, FR, IN, and FP are in balance in the long run, but these are out of equilibrium in the short run. The short-run GNE is out of equilibrium, and the long-run and short-run FR & FP discrepancies are resolved within a year as IG lags in adjusting to the independent variables. According to [51], renewable energy positively impacts inflation and lowers $CO_2$ emissions. The $CO_2$ emissions are considerably detrained by the lag between the 1st and 3rd difference in GNE, IG, FR, IN, FP and $CO_2$ emissions, which are significantly correlated. Even if it changed sign in the second and third lags, the short-run demand situation on IG and IN is strong and good. [52] found that moving government funding to public asset investments increases environmental pollution. The short-run impact of the stringency variable is unfavorable and negligible at the 5% significance level.

**Table 7. Demonstration of ARDL model and Bounds Test results in the long run.**

| Variables | Coeff. | Std. Er. | t-Stat. | Prob. |
|---|---|---|---|---|
| C | 0.515484 | 0.225389 | 2.28709 | 0.0284 |
| $CO_2$ (-1)* | -0.099122 | 0.034554 | -2.868579 | 0.0069 |
| FP_POS** | 0.000276 | 0.002274 | 0.121485 | 0.904 |
| FP_NEG** | -0.007432 | 0.002114 | -3.516251 | 0.0012 |
| FR_POS(-1) | 0.003503 | 0.001059 | 3.308463 | 0.0022 |
| FR_NEG** | -0.003924 | 0.001214 | -3.231284 | 0.0027 |
| GNE_POS** | 0.019343 | 0.021288 | 0.908622 | 0.3698 |
| GNE_NEG** | -0.023527 | 0.016326 | -1.441057 | 0.1585 |
| IN_POS(-1) | -0.040648 | 0.012084 | -3.363829 | 0.0019 |
| IN_NEG(-1) | 0.004608 | 0.007041 | 0.654412 | 0.5171 |
| IG_POS(-1) | -0.006396 | 0.003601 | -1.775947 | 0.0844 |
| IG_NEG** | 0.004888 | 0.002517 | 1.94225 | 0.0602 |
| D(CO2(-1)) | 0.391798 | 0.110493 | 3.545897 | 0.0011 |
| D(FR_POS) | 0.001245 | 0.001187 | 1.048908 | 0.3014 |
| D(FR_POS(-1)) | 0.000478 | 0.001181 | 0.405151 | 0.6878 |
| D(FR_POS(-2)) | 0.002443 | 0.001198 | 2.038732 | 0.0491 |
| D(IN_POS) | -0.008294 | 0.009222 | -0.899348 | 0.3746 |
| D(IN_POS(-1)) | 0.040481 | 0.010917 | 3.708142 | 0.0007 |
| D(IN_NEG) | 0.005169 | 0.009625 | 0.537014 | 0.5947 |
| D(IN_NEG(-1)) | -0.021355 | 0.007756 | -2.753435 | 0.0093 |
| D(IG_POS) | 0.001069 | 0.002684 | 0.398333 | 0.6928 |
| Equation Level | | | | |
| Case 2: Controlled Persistent | | | | |
| Variables | Coeff. | Std. Er. | t-Stat. | Prob. |
| FP_POS | 0.002787 | 0.023363 | 0.119289 | 0.9057 |
| FP_NEG | -0.074979 | 0.022772 | -3.292623 | 0.0023 |
| FR_POS | 0.035337 | 0.011396 | 3.100915 | 0.0038 |
| FR_NEG | -0.039587 | 0.020374 | -1.942995 | 0.0601 |
| GNE_POS | 0.195143 | 0.18493 | 1.055227 | 0.2986 |
| GNE_NEG | -0.237356 | 0.176382 | -1.34569 | 0.1871 |
| IN_POS | -0.410086 | 0.176535 | -2.322966 | 0.0261 |
| IN_NEG | 0.046489 | 0.065756 | 0.70699 | 0.4843 |
| IG_POS | -0.064526 | 0.025837 | -2.497426 | 0.0174 |
| IG_NEG | 0.049314 | 0.031636 | 1.558759 | 0.1281 |
| C | 5.200507 | 1.936903 | 2.684961 | 0.011 |

Note: P-values are significant at 1%, 5%, and 10% levels and are incompatible with t-bounds distribution.

## 4.4 ARDL and NARDL models and F-bound test

Table 9 asserts the presence of a long-run cointegration relationship by validating the bound test estimations for both ARDL and NARDL. The F-Bounds test has been applied to confirm a cointegration relationship of independent and dependent variables in the long run. The F-Bounds test is a statistical test used to determine whether cointegration exists between all the variables in an ARDL model. The F-statistic is then compared to the distribution's upper and lower critical values, which are based on the number of variables in the model and the sample size.

**Table 8. ARDL model error correction regression.**

| Variable | Coefficient | Std. Error | t-Statistic | Prob. |
|---|---|---|---|---|
| $D(CO_2 (-1))$ | 0.391798 | 0.07667 | 5.110184 | 0.0000 |
| D(FR_POS) | 0.001245 | 0.000877 | 1.419047 | 0.1647 |
| D(FR_POS(-1)) | 0.000478 | 0.000836 | 0.572042 | 0.5709 |
| D(FR_POS(-2)) | 0.002443 | 0.00085 | 2.875407 | 0.0068 |
| D(IN_POS) | -0.008294 | 0.005907 | -1.404032 | 0.1691 |
| D(IN_POS(-1)) | 0.040481 | 0.007128 | 5.678907 | 0.0000 |
| D(IN_NEG) | 0.005169 | 0.005828 | 0.886888 | 0.3812 |
| D(IN_NEG(-1)) | -0.021355 | 0.005469 | -3.904627 | 0.0004 |
| D(IG_POS) | 0.001069 | 0.001657 | 0.645024 | 0.5231 |
| CointEq(-1)* | -0.099122 | 0.011924 | -8.313102 | 0 |
| R-squared | 0.819263 | Mean dependent var | | 0.122969 |
| Adjusted R- squared | 0.783901 | SD dependent var | | 0.166027 |
| SE of regression | 0.07718 | Akaike info criterion | | -2.124922 |
| Sum squared resid | 0.27401 | Schwarz criterion | | -1.763252 |
| Log-likelihood | 69.49781 | Hannan-Quinn criter. | | -1.984703 |
| Durbin-Watson stat | 2.039137 | | | |

Note: P-value is significant at 1%, 5% and 10% levels and incompatible with t-Bounds distribution.

Table 10 demonstrates the NARDL model results for all the variables in the short run. The study has applied the NARDL model of order (2, 0, 0, 3, 0, 0, 0, 2, 2, 1, 0) to determine the results of all variables. It is helpful to analyze the short-run consequences of changes in emissions of GNE, IG, FR, IN, and FP using the NARDL model [53]. In the short run, the NARDL model can estimate a shock's immediate and lagged effects on GNE, IG, FR, IN, FP and $CO_2$ emissions.

The results of the NARDL model in the short run will depend on the specific variables and data used in the analysis. The results show that the GNE, IG, FR, IN, and FP effects positively impact $CO_2$ emissions. Inflation (IN), which is large and positive, has a considerable short-run

**Table 9. The f-Bounds Test confirmed a cointegration connection between the variables.**

| Test Statistic | Value | Sign. | I(0) | I(1) |
|---|---|---|---|---|
| | | | Asymptotic: n = 1000 | |
| F-statistic | 4.381827 | 10% | 1.76 | 2.77 |
| k | 10 | 5% | 1.98 | 3.04 |
| | | 2.5% | 2.18 | 3.28 |
| | | 1% | 2.41 | 3.61 |
| Actual Sample Size | 56 | | Finite Sample: n = 60 | |
| | | 10% | -1 | -1 |
| | | 5% | -1 | -1 |
| | | 1% | -1 | -1 |
| | | | Finite Sample: n = 55 | |
| | | 10% | -1 | -1 |
| | | 5% | -1 | -1 |
| | | 1% | -1 | -1 |

Note: Null Hypothesis: No level relationship is found.

**Table 10. Dynamic estimation of the NARDL Model in the short-run.**

| Variable | Coefficient | Std. Error | t-Statistic | Prob.* |
|---|---|---|---|---|
| $CO_2$ (-1) | 1.292677 | 0.109539 | 11.80104 | 0.0000 |
| $CO_2$ (-2) | -0.391798 | 0.110493 | -3.545897 | 0.0011 |
| FP_POS | 0.000276 | 0.002274 | 0.121485 | 0.904 |
| FP_NEG | -0.007432 | 0.002114 | -3.516251 | 0.0012 |
| FR_POS | 0.001245 | 0.001187 | 1.048908 | 0.3014 |
| FR_POS (-1) | 0.002736 | 0.001528 | 1.790837 | 0.082 |
| FR_POS (-2) | 0.001964 | 0.001521 | 1.291718 | 0.2049 |
| FR_POS (-3) | -0.002443 | 0.001198 | -2.038732 | 0.0491 |
| FR_NEG | -0.003924 | 0.001214 | -3.231284 | 0.0027 |
| GNE_POS | 0.019343 | 0.021288 | 0.908622 | 0.3698 |
| GNE_NEG | -0.023527 | 0.016326 | -1.441057 | 0.1585 |
| IN_POS | -0.008294 | 0.009222 | -0.899348 | 0.3746 |
| IN_POS (-1) | 0.008127 | 0.010598 | 0.766862 | 0.4483 |
| IN_POS (-2) | -0.040481 | 0.010917 | -3.708142 | 0.0007 |
| IN_NEG | 0.005169 | 0.009625 | 0.537014 | 0.5947 |
| IN_NEG (-1) | -0.021916 | 0.010485 | -2.090169 | 0.0439 |
| IN_NEG (-2) | 0.021355 | 0.007756 | 2.753435 | 0.0093 |
| IG_POS | 0.001069 | 0.002684 | 0.398333 | 0.6928 |
| IG_POS (-1) | -0.007465 | 0.003048 | -2.449295 | 0.0195 |
| IG_NEG | 0.004888 | 0.002517 | 1.94225 | 0.0602 |
| C | 0.515484 | 0.225389 | 2.28709 | 0.0284 |
| R-squared | 0.999008 | Mean dependent var | | 3.119733 |
| Adjusted R-squared | 0.998442 | SD dependent var | | 2.241374 |
| SE of regression | 0.088481 | Akaike info criterion | | -1.732065 |
| Sum squared resid | 0.27401 | Schwarz criterion | | -0.972558 |
| Log-likelihood | 69.49781 | Hannan-Quinn criter. | | -1.437606 |
| F-statistic | 1762.916 | Durbin-Watson stat | | 2.039137 |
| Prob(F-statistic) | 0 | | | |

Note: The p-values are significant at 1%, 5% and 10%.

impact on rising $CO_2$ emissions. Between 0.03% and 0.97% of China's GDP is used to offset the cost of inflation, with foreigners and investors bearing most of the burden [54]. Although China has agreed on the need to reduce $CO_2$ emissions completely, there are still disparities in regional emissions. Reduced carbon emissions serve the public benefit and reveal a significant positive externality that is challenging to address in the market [11]. Due to China's and India's industrial revolutions, industrial growth (IG) is the primary factor driving the country's rise in $CO_2$ emissions in the next decades [55]. FP and FR impact short-term $CO_2$ emissions and positively correlate with other parameters. Spreading information to fertilizer wholesalers, crop advisors, farmers, and agricultural and environmental authorities should boost BMP consumption [56]. Positive shocks in FP have a positive and substantial coefficient that greatly influences $CO_2$ emissions. An increase in FP causes $CO_2$ emissions to rise, whereas negative shocks to IN usage substantially impact $CO_2$ emissions. A decrease in the usage of renewable energy sources leads to a rise in $CO_2$ emissions, according to the negative IN coefficient. Table 11 reveals that the NARDL model long-run coefficient will converge to equilibrium.

**Table 11. NARDL model long-run and limited constant with no trends.**

| Variable | Coeff. | Std. Er. | t-Stat. | Probability |
|---|---|---|---|---|
| C | 0.515484 | 0.225389 | 2.28709 | 0.0284 |
| CO2 (-1)* | -0.099122 | 0.034554 | -2.868579 | 0.0069 |
| FP_POS** | 0.000276 | 0.002274 | 0.121485 | 0.904 |
| FP_NEG** | -0.007432 | 0.002114 | -3.516251 | 0.0012 |
| FR_POS (-1) | 0.003503 | 0.001059 | 3.308463 | 0.0022 |
| FR_NEG** | -0.003924 | 0.001214 | -3.231284 | 0.0027 |
| GNE_POS** | 0.019343 | 0.021288 | 0.908622 | 0.3698 |
| GNE_NEG** | -0.023527 | 0.016326 | -1.441057 | 0.1585 |
| IN_POS (-1) | -0.040648 | 0.012084 | -3.363829 | 0.0019 |
| IN_NEG (-1) | 0.004608 | 0.007041 | 0.654412 | 0.5171 |
| IG_POS (-1) | -0.006396 | 0.003601 | -1.775947 | 0.0844 |
| IG_NEG** | 0.004888 | 0.002517 | 1.94225 | 0.0602 |
| D(CO2 (-1)) | 0.391798 | 0.110493 | 3.545897 | 0.0011 |
| D(FR_POS) | 0.001245 | 0.001187 | 1.048908 | 0.3014 |
| D(FR_POS(-1)) | 0.000478 | 0.001181 | 0.405151 | 0.6878 |
| D(FR_POS(-2)) | 0.002443 | 0.001198 | 2.038732 | 0.0491 |
| D(IN_POS) | -0.008294 | 0.009222 | -0.899348 | 0.3746 |
| D(IN_POS(-1)) | 0.040481 | 0.010917 | 3.708142 | 0.0007 |
| D(IN_NEG) | 0.005169 | 0.009625 | 0.537014 | 0.5947 |
| D(IN_NEG(-1)) | -0.021355 | 0.007756 | -2.753435 | 0.0093 |
| D(IG_POS) | 0.001069 | 0.002684 | 0.398333 | 0.6928 |
| Case 2: Constrained Persistent | | | | |
| Variable | Coefficient | Std. Error | t-Statistic | Prob. |
| FP_POS | 0.002787 | 0.023363 | 0.119289 | 0.9057 |
| FP_NEG | -0.074979 | 0.022772 | -3.292623 | 0.0023 |
| FR_POS | 0.035337 | 0.011396 | 3.100915 | 0.0038 |
| FR_NEG | -0.039587 | 0.020374 | -1.942995 | 0.0601 |
| GNE_POS | 0.195143 | 0.18493 | 1.055227 | 0.2986 |
| GNE_NEG | -0.237356 | 0.176382 | -1.34569 | 0.1871 |
| IN_POS | -0.410086 | 0.176535 | -2.322966 | 0.0261 |
| IN_NEG | 0.046489 | 0.065756 | 0.70699 | 0.4843 |
| IG_POS | -0.064526 | 0.025837 | -2.497426 | 0.0174 |
| IG_NEG | 0.049314 | 0.031636 | 1.558759 | 0.1281 |
| C | 5.200507 | 1.936903 | 2.684961 | 0.011 |

Note: The p-values are significant at 1%, 5% and 10% levels and are compatible with t-bounds distribution.

The results demonstrate that the positive shocks to GNE and IG meaningfully affect fossil fuel byproducts' high likelihood worth and negative coefficient esteem. China's $CO_2$ emanations are increasing over the long haul because the nation is making an ever-increasing number of farming items [57]. A positive change in IG affects $CO_2$ discharges, showing that contamination in the climate decreases as IG improves. It is another significant component affecting fossil fuel byproducts when a nation's funds deteriorate; because less cash is in the in the evolving hands, and pollution levels go up. Conversely, negative shocks to IG show a reasonable connection with fossil fuel byproducts.

**Table 12. F-Bounds test to assert the cointegration connections.**

| Test Statistic | Value | Signif. | I(0) | I(1) |
|---|---|---|---|---|
|  |  |  | Asymptotic: n = 1000 |  |
| F-statistic | 4.381827 | 10% | 1.76 | 2.77 |
| k | 10 | 5% | 1.98 | 3.04 |
|  |  | 2.5% | 2.18 | 3.28 |
|  |  | 1% | 2.41 | 3.61 |
| Actual Sample Size | 56 |  | Finite Sample: n = 60 |  |
|  |  | 10% | -1 | -1 |
|  |  | 5% | -1 | -1 |
|  |  | 1% | -1 | -1 |
|  |  |  | Finite Sample: n = 55 |  |
|  |  | 10% | -1 | -1 |
|  |  | 5% | -1 | -1 |
|  |  | 1% | -1 | -1 |

Note: Null Hypothesis: There is not any relationship found on any level

## 4.5 NARDL bounds test

Table 12 demonstrates the F-Bounds test to assert the cointegration connection in the short- and long-run. The NARDL model with the Bounds test has been used in econometric analysis to determine the relationship between study macroeconomic variables and $CO_2$ emissions. Once the NARDL model is estimated, the study will conduct the F-Bounds test to assess the existence of cointegration between the variables. The F-Bounds test involves calculating the F-statistics upper and lower bounds based on macroeconomic variables' impact on $CO_2$ emissions.

Industrial revolutions have diverse implications for achieving net-zero carbon emissions [58]. The study assessed the cointegration connection between the variables and gained insights into their long-run relationships by calculating the upper and lower bounds of the F-statistic. China's industrial sector investment volume increases $CO_2$ emissions [34].

## 4.6 CUSUM and CUSUM of square graphs

Fig 2 demonstrates the CUSUM and CUSUMSQ of the square model's stability. The CUSUM test involves calculating a sequence of test statistics that represent the cumulative sum of the residuals from the regression model, and the results of test statistics have been plotted against the sample size and a critical value. The CUSUMSQ test involves calculating a sequence of test statistics representing the cumulative sum of the squared residuals from the regression model. The parameter stability is evaluated using the cumulative sum of recursive residuals (CUSUM) and cumulative sum of squares (CUSUMSQ) tests [59]. The study used the CUSUM and CUSUM of Squares (CUSUMSQ) tests to discover the NARDL model residuals and the link between GNE, IN, IG, FP, and $CO_2$ emissions. Study test data were plotted against sample size and a critical value. CUSUM and CUSUM of Square tests identify time series mean shifts.

The graphs indicate a significant change in the mean of FP, GNE, IN, and IG related to $CO_2$ emissions, and it suggests a potential causal relationship between FP, GNE, IN, IG and $CO_2$ emissions. Table 13 demonstrates the diagnostic inspection test for verification of the ARDL and NARDL model results. The ARDL and NARDL models have assumed that the errors are normally distributed, and deviations from normality can lead to biased coefficient

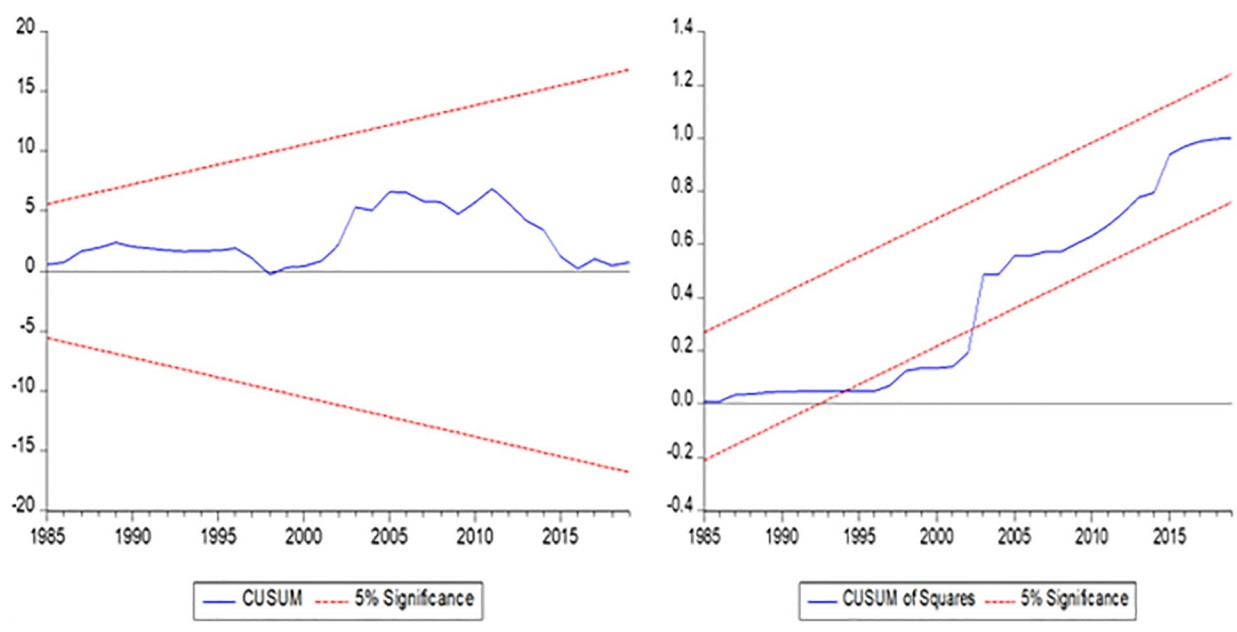

**Fig 2. The CUSUM and CUSUM of squares test results of macroeconomic variables. Source:** Author's illustration.

estimates [60]. The heteroskedasticity-hearty Breusch-Agnostic test is more reliable than the "wild bootstrap form" of the standardized unique Breusch-Agnostic test [61]. The diagnostic tests have been applied to identify potential problems with the ARDL and NARDL models and improve the overall performance of all the variables.

The Jarque-Bera test's probability values are more than 5%, which shows that the null hypothesis of normality is not disproved. The evidence for a U-shaped connection between the rate of industrial expansion and $CO_2$ emission was supported by both short-run and long-run regression parameters [62]. The hidden ARDL and NARDL models' relapses match very well; both are critical at the 1% level worldwide. These models passed the Lagrange Multipliers (LM) test for heteroskedasticity, the Jarque-Bera test, the Breusch-Agnostic Godfrey test, and the RESET test. The consumption of fertilizers and fossil fuels consumption increased environmental pollution [63].

## 4.7 Estimation of a ramsey RESET model

Table 14 demonstrates the RESET test confirming the results of the ARDL and NARDL models. The Ramsey RESET test is a diagnostic test applied to determine a nonlinear regression

**Table 13. Diagnostic test inspection.**

| Diagnostic tests | Problem | p-value | Result |
|---|---|---|---|
| LM | Serial Correlation | 0.632 | Inconsistent temporal correlations |
| Jarqure-Bera | Normality | 0.7361 | The distribution of residuals is normal |
| Breusch—Pagan—Godfrey | Heteroscedasticity | 0.8967 | In the absence of heteroscedasticity |
| Ramsey RESET test | Model requirement | 0.1500 | The model has a valid specification |
| CUSTOM | Stability | - | Model is steady |
| CUSUMSQ | Stability | - | The model needs to be more steady. |

**Table 14. Ramsey RESET test with F-test summary.**

|  | **Value** | **df** | **Probability** |
|---|---|---|---|
| t-stat. | 0.403682 | 43 | 0.6884 |
| F-stat. | 0.162959 | (1, 43) | 0.6884 |
| Ramsey RESET Test | | | |
| Misplaced Variables: Quadrangles of fitted values | | | |
|  | Value | df | Probability |
| t-stat. | 1.472796 | 34 | 0.1500 |
| F-stat. | 2.169130 | (1, 34) | 0.1500 |
| F-test summary | | | |
|  | Sum of Sq. | df | Mean Sq. |
| Test SSR | 0.016433 | 1 | 0.016433 |
| Restricted SSR | 0.274010 | 35 | 0.007829 |
| Unrestricted SSR | 0.257577 | 34 | 0.007576 |

*Note: P-values and any subsequent tests do not account for the model

between the independent and dependent variables. The Ramsey RESET test is also a diagnostic tool used with other diagnostic tests to ensure the model is correctly specified.

It estimates the regression model of $CO_2$ emissions as a function of Gross National Expenditures (GNE), Inflation and Industrial Growth. The results of the Ramsey RESET test have been interpreted in conjunction with other diagnostic tests and considerations, and the residuals for patterns and testing for multi-collinearity among the regressors.

## 4.8 Granger causality test

Table 15 demonstrates the Granger causality scales between the components after the cointegration coefficients. The Granger causality test is a statistical technique used to determine whether one variable is used to predict another variable. The study has applied the Granger causality test to analyze the time series data for stationarity.

The results reveal that the positive shocks of GNE to $CO_2$ emissions are correlated in both directions; negative shocks in IG, FP, and GNE Granger cause $CO_2$ emissions. A unidirectional association between IG usage and FR confirms China's $CO_2$ emissions. $CO_2$ emissions have positive shocks in IG, negative shocks in IN, and negative shocks in FP, which are all brought on by Granger. To validate the GNE theory, there is also a unidirectional association between FR, IN, GNE, and FP. Table 16 demonstrates the Granger causality test result for the effectiveness of all variables on $CO_2$ emissions.

**Table 15. Demonstrates that the Granger causality test estimations.**

| Lag | LogL | LR | FPE | AIC | SC | HQ |
|---|---|---|---|---|---|---|
| 0 | -862.535 | NA | 3763033. | 32.16797 | 32.38897 | 32.25320 |
| 1 | -772.984 | 155.8852* | 522374.4* | 30.18460* | 31.73159* | 30.78121* |
| 2 | -749.837 | 35.15014 | 883092.4 | 30.66061 | 33.53359 | 31.76861 |
| 3 | -718.391 | 40.76282 | 1191207. | 30.82929 | 35.02826 | 32.44867 |
| 4 | -687.907 | 32.74183 | 1912063. | 31.03360 | 36.55855 | 33.16436 |
| 5 | -643.984 | 37.41628 | 2342967. | 30.74014 | 37.59108 | 33.38228 |

Note: The p-values are significant at the 5% level.

**Table 16. Granger causality test result for the effectiveness of $CO_2$ emissions.**

| Variables | $CO_2$ | FP | FR | GNE | IN | IG |
|---|---|---|---|---|---|---|
| CO2 | - | ≠ | ≠ | ≠ | → | ≠ |
| FP | ≠ | - | ≠ | ≠ | ≠ | ↔ |
| FR | ≠ | ≠ | - | → | ≠ | ≠ |
| GNE | ≠ | ≠ | → | - | ≠ | ≠ |
| IN | → | ≠ | ≠ | ≠ | - | → |
| IG | ≠ | ↔ | ≠ | ≠ | → | - |

The results of the Granger causality test for the association between GNE, IG, and FR are intriguing. These results show that neither the Granger causality estimation from GNE to IN nor IN to IG is statistically significant. The NARDL model reveals that changes in energy usage, fertilizer use, and agricultural carbon emission lead to changes in cereal food production, both positively and negatively (Koondhar et al., 2021) [64]. The findings demonstrate that GNE, FP, IG, and FR growth will only increase $CO_2$ emissions. Table 17 shows the impulse response function of $CO_2$ emissions to confirm the relationship validity between the study variables. The impulse response function indicates that economic activity raises $CO_2$ emissions mathematically.

The results demonstrate that $CO_2$ emissions have an inverse connection with negative shocks to GNE and IG. The positive shocks to IN decrease $CO_2$ emissions, whereas an increase in IG and GNE also increases the $CO_2$ emissions. China's $CO_2$ emissions rise in response to positive shocks to FP and FR. Within a decade, green finance will significantly reduce fertilizer use and agricultural carbon emissions [65]. $CO_2$ emissions will fall along with an increase in FP, and China's GNE and IG will also significantly influence $CO_2$ emissions.

**Table 17. Demonstrates the Impulse Response Function of $CO_2$ emissions.**

| Period | $CO_2$ | FP | FR | GNE | IN | IG |
|---|---|---|---|---|---|---|
| 1 | 0.121569 | -1.919689 | 0.152964 | 0.031073 | 0.745434 | 1.819074 |
| | (0.01129) | (1.21521) | (2.17342) | (0.22161) | (0.39695) | (0.85565) |
| 2 | 0.188772 | -2.10732 | 3.916587 | -0.055897 | 0.828937 | 1.437827 |
| | (0.02335) | (1.44787) | (3.15290) | (0.26517) | (0.49763) | (0.94781) |
| 3 | 0.219088 | -1.672895 | 6.537434 | -0.21959 | 0.500467 | 0.421282 |
| | (0.03848) | (1.66204) | (4.13901) | (0.29140) | (0.56053) | (0.97650) |
| 4 | 0.229610 | -1.50194 | 7.400607 | -0.27025 | 0.065907 | -0.224304 |
| | (0.05214) | (1.64835) | (4.92364) | (0.28970) | (0.53288) | (0.78256) |
| 5 | 0.232278 | -1.559095 | 7.291290 | -0.237078 | -0.251749 | -0.371268 |
| | (0.06267) | (1.50702) | (5.48087) | (0.26608) | (0.44652) | (0.60772) |
| 6 | 0.233197 | -1.715131 | 6.720930 | -0.176879 | -0.385551 | -0.279988 |
| | (0.07047) | (1.35376) | (5.85169) | (0.23532) | (0.36024) | (0.47030) |
| 7 | 0.233808 | -1.840843 | 5.980471 | -0.120616 | -0.394174 | -0.199189 |
| | (0.07680) | (1.25672) | (6.13927) | (0.20858) | (0.31299) | (0.35443) |
| 8 | 0.233510 | -1.881189 | 5.216439 | -0.0757 | -0.361416 | -0.210833 |
| | (0.08283) | (1.23309) | (6.43372) | (0.19078) | (0.30306) | (0.30008) |
| 9 | 0.231622 | -1.854638 | 4.470014 | -0.040066 | -0.338267 | -0.270509 |
| | (0.08912) | (1.25459) | (6.77615) | (0.18177) | (0.31271) | (0.28493) |
| 10 | 0.227991 | -1.800091 | 3.728429 | -0.010646 | -0.335753 | -0.317435 |
| | (0.09569) | (1.29328) | (7.16688) | (0.17948) | (0.32698) | (0.28122) |

Table 18. Variance decomposition analysis of $CO_2$ emissions.

| Period | SE. | CO₂ | FP | FR | GNE | IN | IG |
|--------|-----|-----|-----|-----|-----|-----|-----|
| 1 | 0.121569 | 100.0000 | 0.000000 | 0.000000 | 0.000000 | 0.000000 | 0.000000 |
| 2 | 0.228267 | 96.75268 | 0.927161 | 1.827193 | 0.281943 | 0.186056 | 0.024971 |
| 3 | 0.326678 | 92.21814 | 2.314773 | 4.479798 | 0.341747 | 0.100969 | 0.544575 |
| 4 | 0.415055 | 87.73077 | 3.496304 | 6.692945 | 0.484751 | 0.237772 | 1.357455 |
| 5 | 0.494634 | 83.82469 | 4.054854 | 8.837601 | 0.866436 | 0.405086 | 2.011330 |
| 6 | 0.568744 | 80.21417 | 4.091648 | 11.31704 | 1.484600 | 0.506660 | 2.385883 |
| 7 | 0.640219 | 76.64041 | 3.876098 | 14.19197 | 2.176528 | 0.545764 | 2.569225 |
| 8 | 0.709977 | 73.13741 | 3.599171 | 17.27997 | 2.778724 | 0.542492 | 2.662241 |
| 9 | 0.777641 | 69.83505 | 3.336780 | 20.37284 | 3.229739 | 0.511527 | 2.714073 |
| 10 | 0.842661 | 66.79410 | 3.097934 | 23.36399 | 3.542927 | 0.464348 | 2.736700 |

## 4.9 Variance decomposition analysis

Variance decomposition analysis has been applied to decompose the variance of the macroeconomic research variables and $CO_2$ emissions. A quarter of all human $CO_2$ emissions are brought on by land usage and agricultural output [66]. There are positive shocks of IG, FR, and GNE to $CO_2$ emissions, but FP has a negative shock FP to $CO_2$ emissions. Table 18 demonstrates the variance decomposition analysis (VDA) of $CO_2$ emissions.

The results demonstrate that FP, GNE, IG and IN significantly impact $CO_2$ emissions. The GNE is the most important variable, and policymakers might focus on promoting economic growth through sustainable and low-carbon technologies while reducing energy consumption and improving energy efficiency.

## 4.10 Discussions

The findings show that greater gross national expenditures (GNE) correspond to greater economic activity, which increases $CO_2$ emissions. Currently, no regulatory interactions link increasing economic development and reducing carbon dioxide emissions at the national level [67]. Most economic activity depends on energy usage, normally produced using fossil fuels that emit $CO_2$. Higher inflation rates cause lower $CO_2$ emissions. Using and manufacturing fertilizers have expanded, increasing greenhouse gas emissions. In addition, excessive fertilizer usage can result in soil deterioration and nutrient loss, decreasing soil carbon sequestration and raising $CO_2$ emissions. Using waste instead of fossil fuels may reduce $CO_2$ emissions [68]. Finally, GNE, inflation, fertilizer use, and industrial growth greatly influence $CO_2$ emissions because industrial growth in China often entails expanding manufacturing and output at a high level, increasing $CO_2$ emissions. The excess use of fertilizers to increase agricultural production causes environmental pollution, especially increasing $CO_2$ emissions [21]. The development of targeted industry-based greenhouse gas reduction strategies. The top-down analysis allows the assessment of tourism as a sector within the wider economy [69]. For every 1% increase in tourism demand, foreign direct investment (FDI) has a 0.22% positive effect and a 0.54% negative effect. In China, an unbalanced correlation exists between foreign direct investment (FDI) and tourism, seemingly stemming from a unidirectional causal relationship [70]. While inflation helps to lower $CO_2$ emissions, it also lowers industrial growth when people's purchasing power declines due to rising inflation. The confirmation of cointegration among the variables and both short- and long-run regression parameters indicated evidence of a U-shaped association between the level of industrial growth and $CO_2$ emissions [62].

## 5. Conclusions and recommendations

The rapid economic growth and development have led to an increased reliance on fossil fuels, particularly oil, which has significantly increased $CO_2$ emissions in China. The study has been researched to determine the influences of gross national expenditures (GNE), inflation (IN), fertilizer consumption (FP), and industrial growth (IG) on $CO_2$ emissions using time series data from 1960 to 2022. The methodology has applied ARDL and NARDL models to analyse the short- and long-run data. The Granger causality, IRF, and VDA are also utilized to determine the relationship between the GNE, IN, FP, IG, and $CO_2$ emissions. The F-bound test has been used to confirm the long-run cointegration of all the variables. The results demonstrate that momentary $CO_2$ emissions have a solid and unfavorable relationship with GNE and IG. The Granger causality results show that FR, IG, FP, and GNE significantly impact $CO_2$ emissions. The relationship between inflation and GDP is positive, but FP, GNE, and IG hurt $CO_2$ emissions. Additionally, it is important to consider other factors that may influence $CO_2$ emissions, such as population growth, economic conditions, and energy policies. The IRF found negative shocks to GNE and $CO_2$ emissions but positive shocks to GNE, IG, and $CO_2$ emissions. A negative shock to GNE is thought to result in a reduction in $CO_2$ emissions, according to the IRF research, which also revealed negative shocks to GNE and $CO_2$ emissions. A positive shock to GNE or IG is thought to boost $CO_2$ emissions, according to the IRF analysis, which also discovered positive shocks to GNE, IG, and $CO_2$ emissions. The VDA shows negative IG, GNE, FR, and $CO_2$ emissions shocks. The VDA analysis reveals negative IG, GNE, FR, and $CO_2$ emissions shocks. The possible links between the four variables investigated in the VDA and other variables could affect the outcome variables. The impact of FR on $CO_2$ emissions has altered due to institutional sufficiency. The FP and IN positively impact $CO_2$ emissions, and a 1% increase in GNE will also increase $CO_2$ emissions. Higher industrial growth (IG) has an unequal impact on $CO_2$ emissions. The impact of industrial growth on $CO_2$ emissions may depend on the energy intensity of the industrial sector, the use of renewable or fossil fuel-based energy sources, the efficiency of production processes, the level of technology and innovation, and the environmental regulations and policies in place. Because of the rise in fossil fuel byproducts caused by FR and FP, expansion (IN) also exhibits unbalanced behavior. FP, IG, and GNE are key drivers of the increase in $CO_2$ emissions in China, as the burning of fossil fuels is the largest contributor to anthropogenic $CO_2$ emissions. As the demand for energy continues to grow, particularly in developing countries, the use of fossil fuels is likely to increase, leading to further increases in $CO_2$ emissions. China is the world's largest emitter of $CO_2$ emissions, accounting for over a quarter of global emissions. FP, IG, and GNE are identified as key drivers of the increase in $CO_2$ emissions in China. $CO_2$ emissions increase global temperatures, precipitation patterns, extreme weather events, and ocean levels. There is a need to shift towards cleaner and more sustainable sources of energy, such as renewable energy sources like solar, wind, and hydropower.

### 5.1 Future research suggestions

Future research can investigate how industrial growth affects emissions across different industries or countries and identify policies or practices that can help reduce emissions associated with industrial growth. Overall, these research suggestions could help better understand the complex relationships between economic growth, agricultural practices, industrial development, and $CO_2$ emissions and identify strategies to reduce emissions and mitigate the impacts of climate change.

## Author Contributions

**Conceptualization:** Abdullah Addas, Liaqat Ali Waseem, Syed Ali Asad Naqvi, Muneeb Ahmad.

**Data curation:** Abdullah Addas, Syed Ali Asad Naqvi, Muneeb Ahmad, Kashif Sharif.

**Formal analysis:** Syed Ali Asad Naqvi, Muneeb Ahmad, Kashif Sharif.

**Investigation:** Dan Zheng, Syed Ali Asad Naqvi, Kashif Sharif.

**Methodology:** Abdullah Addas, Syed Ali Asad Naqvi, Kashif Sharif.

**Project administration:** Dan Zheng.

**Software:** Dan Zheng, Liaqat Ali Waseem, Muneeb Ahmad.

**Supervision:** Muneeb Ahmad.

**Validation:** Abdullah Addas, Liaqat Ali Waseem, Syed Ali Asad Naqvi, Kashif Sharif.

**Visualization:** Dan Zheng, Muneeb Ahmad.

**Writing – original draft:** Dan Zheng, Abdullah Addas, Liaqat Ali Waseem, Kashif Sharif.

**Writing – review & editing:** Dan Zheng, Abdullah Addas.

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
