## [Decision Letter · Decision Letter 0]

18 Sep 2023

PONE-D-23-18102Analyzing the Influence of Economic Factors on CO2 Emissions in China: A NARDL ApproachPLOS ONE

Dear Dr. Dr. Ahmad,

Thank you for submitting your manuscript to PLOS ONE. After careful consideration, we feel that it has merit but does not fully meet PLOS ONE’s publication criteria as it currently stands. Therefore, we invite you to submit a revised version of the manuscript that addresses the points raised during the review process.

Please submit your revised manuscript by Nov 02 2023 11:59PM If you will need more time than this to complete your revisions, please reply to this message or contact the journal office at plosone@plos.org. Please include the following items when submitting your revised manuscript:A rebuttal letter that responds to each point raised by the academic editor and reviewer(s). You should upload this letter as a separate file labeled 'Response to Reviewers'.A marked-up copy of your manuscript that highlights changes made to the original version. You should upload this as a separate file labeled 'Revised Manuscript with Track Changes'.An unmarked version of your revised paper without tracked changes. You should upload this as a separate file labeled 'Manuscript'.

We look forward to receiving your revised manuscript.

Kind regards,

Nikeel Nishkar Kumar

Academic Editor

PLOS ONE

Journal Requirements:

Reviewers' comments:

Reviewer's Responses to Questions

**Comments to the Author**

1. Is the manuscript technically sound, and do the data support the conclusions?

Reviewer #1: Partly

Reviewer #2: Yes

2. Has the statistical analysis been performed appropriately and rigorously? 

Reviewer #1: No

Reviewer #2: Yes

3. Have the authors made all data underlying the findings in their manuscript fully available?

Reviewer #1: Yes

Reviewer #2: Yes

4. Is the manuscript presented in an intelligible fashion and written in standard English?

Reviewer #1: Yes

Reviewer #2: Yes

5. Review Comments to the Author

Reviewer #1: I would like to thank the authors for compiling the article entitled: Analyzing the Influence of Economic Factors on CO2 Emissions in China: A NARDL Approach. Presently, this study has not met the criteria for examining the relationship between vigorous reporting of results and the lack of novelty for the paper. Please find the comments suitable for restructuring the paper.

1. The first sentence in the abstract: ..... The study described examines..... please change the wording.

2. Contrary to employing the NARDL approach, justify your reason based on previous research why the NARDL approach is popular. I believe there are other econometric model which can find the same relationship.

3. Please strengthen the motivation of the paper in terms of the evidence gap and the practical knowledge gap.

4. Justify your claims about why the study is important to China.

5. Expand on the research findings and what other dimensions have been studied in the paper. has the current study expanded on another related research.

6. The literature seems to be too broad. please use thematic literature review, include sub themes in the literature.

7. in the literature section only 2 country specific examples are use. I would suggest to include other country examples.

other minor comments:

multiple text background highlighted in yellow. Please refer to journals publication criteria.

Figure 1 should fit in a single page.

I look forward for an improved version of this paper soon.

All the Best

Reviewer #2: 1) Add novelty in the Introduction part and clearly indicate the contribution of the study to the contemporary literature.

2) add more recent articles in the literature part

3) Add theoretical framework of the study in a different section.

4) Add the different section on policy implications of the study for the policy makers and what are the future areas of the research?

6. PLOS authors have the option to publish the peer review history of their article (what does this mean?). If published, this will include your full peer review and any attached files.

Reviewer #1: No

Reviewer #2: No

---

## [Author Response · Author response to Decision Letter 0]

24 Oct 2023

Authors have added all the required in formation and changes in the revised manuscript, we have also attached reviewer response letter in attachments.

---

## [Decision Letter · Decision Letter 1]

1 Dec 2023

PONE-D-23-18102R1The Hidden Costs of Inflation: A Critical Analysis of Industrial Development and Environmental ConsequencesPLOS ONE

Dear Dr. Ahmad,

Thank you for submitting your manuscript to PLOS ONE. After careful consideration, we feel that it has merit but does not fully meet PLOS ONE’s publication criteria as it currently stands. Therefore, we invite you to submit a revised version of the manuscript that addresses the points raised during the review process. Please submit your revised manuscript by Jan 15 2024 11:59PM. If you will need more time than this to complete your revisions, please reply to this message or contact the journal office at plosone@plos.org. Please include the following items when submitting your revised manuscript:A rebuttal letter that responds to each point raised by the academic editor and reviewer(s). You should upload this letter as a separate file labeled 'Response to Reviewers'.A marked-up copy of your manuscript that highlights changes made to the original version. You should upload this as a separate file labeled 'Revised Manuscript with Track Changes'.An unmarked version of your revised paper without tracked changes. You should upload this as a separate file labeled 'Manuscript'.If applicable, we recommend that you deposit your laboratory protocols in protocols.io to enhance the reproducibility of your results. Protocols.io assigns your protocol its own identifier (DOI) so that it can be cited independently in the future. For instructions see: https://journals.plos.org/plosone/s/submission-guidelines#loc-laboratory-protocols. Additionally, PLOS ONE offers an option for publishing peer-reviewed Lab Protocol articles, which describe protocols hosted on protocols.io. Read more information on sharing protocols at https://plos.org/protocols?utm_medium=editorial-email&utm_source=authorletters&utm_campaign=protocols.

We look forward to receiving your revised manuscript.

Kind regards,

Nikeel Nishkar Kumar

Academic Editor

PLOS ONE

Journal Requirements:

Reviewers' comments:

Reviewer's Responses to Questions

**Comments to the Author**

1. If the authors have adequately addressed your comments raised in a previous round of review and you feel that this manuscript is now acceptable for publication, you may indicate that here to bypass the “Comments to the Author” section, enter your conflict of interest statement in the “Confidential to Editor” section, and submit your "Accept" recommendation.

Reviewer #1: All comments have been addressed

2. Is the manuscript technically sound, and do the data support the conclusions?

Reviewer #1: Yes

3. Has the statistical analysis been performed appropriately and rigorously? 

Reviewer #1: Yes

4. Have the authors made all data underlying the findings in their manuscript fully available?

Reviewer #1: Yes

5. Is the manuscript presented in an intelligible fashion and written in standard English?

Reviewer #1: Yes

6. Review Comments to the Author

Reviewer #1: Dear authors.

Thank you for the resubmission . This paper has significantly improved over the last version.

I would suggest to incorporate papers from Plos one and include in the literature review section.

Also use this research paper to guide and replenish overall novetly of the paper especially the methodology section. https://doi.org/10.1080/19407963.2022.2151605.

Thanks you.

7. PLOS authors have the option to publish the peer review history of their article (what does this mean?). If published, this will include your full peer review and any attached files.

Reviewer #1: No

---

## [Editor Report · Decision Letter 2]

4 Jan 2024

The Hidden Costs of Inflation: A Critical Analysis of Industrial Development and Environmental Consequences

PONE-D-23-18102R2

Dear Dr. Ahmad,

We’re pleased to inform you that your manuscript has been judged scientifically suitable for publication and will be formally accepted for publication once it meets all outstanding technical requirements.

Kind regards,

Nikeel Nishkar Kumar

Academic Editor

PLOS ONE
---

## [Editor Report · Acceptance letter]

22 May 2024

PONE-D-23-18102R2 

PLOS ONE

Dear Dr. Ahmad, 

I'm pleased to inform you that your manuscript has been deemed suitable for publication in PLOS ONE. Congratulations! Your manuscript is now being handed over to our production team.

Kind regards, 

on behalf of

Dr. Nikeel Nishkar Kumar 

Academic Editor

PLOS ONE